# Subcellular Compartmentalization of Glucose Mediated Insulin Secretion

**DOI:** 10.3390/cells14030198

**Published:** 2025-01-29

**Authors:** Zhongying Wang, Tatyana Gurlo, Leslie S. Satin, Scott E. Fraser, Peter C. Butler

**Affiliations:** 1Translational Imaging Center, Michelson Center for Convergent Bioscience, University of Southern California, Los Angeles, CA 90089, USA; zhongyingw7@gmail.com (Z.W.); sfraser@provost.usc.edu (S.E.F.); 2Dana and David Dornsife College of Letters, Arts and Sciences, University of Southern California, Los Angeles, CA 90089, USA; 3Larry L. Hillblom Islet Research Center, David Geffen School of Medicine at UCLA, Los Angeles, CA 90089, USA; tgurlo@g.ucla.edu; 4Brehm Diabetes Center, Caswell Diabetes Institute, Department of Pharmacology, University of Michigan, Ann Arbor, MI 38105, USA; lsatin@umich.edu

**Keywords:** compartmentalization, submembrane, beta cells, K_ATP_ channel, mitochondria, oxidative phosphorylation, insulin, pulsatile secretion, calcium, ATP

## Abstract

Regulation of blood glucose levels depends on the property of beta cells to couple glucose sensing with insulin secretion. This is accomplished by the concentration-dependent flux of glucose through glycolysis and oxidative phosphorylation, generating ATP. The resulting rise in cytosolic ATP/ADP inhibits K_ATP_ channels, inducing membrane depolarization and Ca^2+^ influx, which prompts insulin secretion. Evidence suggests that this coupling of glucose sensing with insulin secretion may be compartmentalized in the submembrane regions of the beta cell. We investigated the subcellular responses of key components involved in this coupling and found mitochondria in the submembrane zone, some tethered to the cytoskeleton near capillaries. Using Fluorescent Lifetime Imaging Microscopy (FLIM), we observed that submembrane mitochondria were the fastest to respond to glucose. In the most glucose-responsive beta cells, glucose triggers rapid, localized submembrane increases in ATP and Ca^2+^ as synchronized ~4-min oscillations, consistent with pulsatile insulin release after meals. These findings are consistent with the hypothesis that glucose sensing is coupled with insulin secretion in the submembrane zone of beta cells. This zonal adaptation would enhance both the speed and energy efficiency of beta cell responses to glucose, as only a subset of the most accessible mitochondria would be required to trigger insulin secretion.

## 1. Introduction

In health, an increase in blood glucose concentration is sensed metabolically and induces a rapid increase in insulin secretion by pancreatic beta cells. In this simple metabolic sensing model, glucose enters beta cells through membrane glucose transporter proteins and is metabolized to pyruvate at a rate proportionate to its concentration since the Km of glucokinase, the rate-limiting enzyme of glycolysis in beta cells, is in the mid-physiological range [1]. This pyruvate is taken up by mitochondria and oxidized by oxidative phosphorylation (OxPhos), generating ATP that, in turn, inhibits K_ATP_ channels in the cell membrane, resulting in cell membrane depolarization, the entry of Ca^2+^ through voltage-gated Ca^2+^ channels, and the Ca^2+^–dependent release of insulin from the docked/primed insulin secretory granule pool near the cell membrane [2,3]. This canonical model for coupling glucose sensing with insulin secretion was established by studies of normal physiology as well as of monozygotic forms of diabetes [4].

Using Fluorescent Lifetime Imaging Microscopy (FLIM) to monitor the metabolism of living beta cells, we noted regional differences, with only some of the mitochondria recruited to active OxPhos in response to glucose stimulation [5]. Other components of the machinery that couples glucose sensing with insulin secretion appear to be clustered at the cell membrane [6]; for example, ATP production [7], glucose transporters [8], and K_ATP_ channels have each been reported as clustered in subregions of the membrane. Other studies have inferred the compartmentalization of glycolytic enzymes, such as pyruvate kinase [9], at the beta cell membrane.

Co-clustering of the components of the glucose sensing and insulin secretion machinery at the cell membrane would offer several advantages; for example, it would facilitate the efficiency and speed of metabolism-secretion coupling. ATP elevation would be greatest near the submembrane localized mitochondria, locally providing a high concentration of ATP to the K_ATP_ channels. This would not only rapidly trigger membrane depolarization but also provide the energy for ion pumping and membrane repolarization and prime the membrane-adjacent secretory granules for docking. Local Ca^2+^ influx upon membrane depolarization would result in a local Ca^2+^ concentration elevation, not only triggering the secretion of the primed/docked secretory granules but also promoting the submembrane mitochondria towards OxPhos by activating the Ca^2+^–dependent dehydrogenases of the TCA cycle. These spatial relationships would naturally lead to a robust coordination of OxPhos, ATP generation and consumption, membrane depolarization and repolarization, Ca^2+^ influx, and OxPhos; such coordination has been elegantly modeled to explain the pulsatile nature of insulin secretion [10,11].

In the present study, we tested the hypothesis that the coupling of glucose sensing with insulin secretion by beta cells is compartmentalized in the submembrane zone.

## 2. Materials and Methods

### 2.1. Cells

INS832/13 cells, made available by Dr. Christopher Newgard (Duke University, Durham, NC, USA) [12], were cultured in RPMI 1640 medium containing 11 mM glucose, 5% heat-inactivated fetal bovine serum, 100 Units/mL penicillin, 100 µg/mL streptomycin, 10 mM HEPES, 1 mM sodium pyruvate, and 50 µM β-mercaptoethanol. The cells were maintained at 37 °C in a humidified atmosphere with 5% CO_2_.

INS832/13 cells were seeded on chambered cover glass (Ibidi USA, Fitchburg, WI, USA, Cat#80826-G500) precoated with Poly-L-lysine (Sigma-Aldrich, St. Louis, MO, USA. Cat#P7280). Poly-L-lysine was diluted in PBS to a concentration of 0.1 mg/mL and applied to the culture surface for 24 h at 4 °C, then rinsed off with PBS. Cells were plated at a density of 70,000 cells per cm^2^ and cultured for 3 to 4 days to reach 70% confluency before imaging.

### 2.2. Animals

FVB/NJ mice were procured from the Charles River Laboratory. The mice were housed in the vivarium of the University of California, Los Angeles, and the University of Southern California. Animal studies at the University of California, Los Angeles, adhered to the guidelines of the UCLA Office of Animal Research Oversight, with ethical approval from the Research Safety and Animal Welfare Administration at UCLA (ARC#2004-119 and ARC#2004-148). Animal studies at the University of California, Los Angeles, adhered to the guidelines of the IACUC board of the University of Southern California (protocol 21220).

### 2.3. Mouse Single Islet Cells

Pancreatic islets were isolated from 9- to 10-week-old male mice after the digestion of the pancreas with collagenase for 17 min in a 37 °C water bath and further handpicked as previously described by Wang et al. [5]. The islets were then washed with PBS, pelleted, and resuspended in 2 mL of ice-cold Accutase cell detachment solution (Innovative Cell Technologies, San Diego, CA, USA. Cat#AT-104). Thereafter, we incubated the islets at 37 °C for 15 min. An equal volume of tissue culture medium (TCM, RPMI with 11 mM glucose supplemented with 100 IU/mL penicillin/streptomycin and 10% fetal bovine serum) was added before centrifugation for 3 min at 300 g and resuspension in TCM for plating on a Ibidi-slide 8 well with grid (Ibidi USA, Inc., Fitchburg, WI, USA. Cat#80826-G500) precoated with Poly-L-Lysine (Sigma-Aldrich, St. Louis, MO, USA. Cat#P6407) for 24 h at 4 °C, followed by Laminin (Life Technologies, Carlsbad, CA, USA. Cat#23017015) for 24 h at 4 °C as described by Wang et al. [5]. Dispersed islet cells were plated at a density of 30,000 per well, and imaging was undertaken 2 to 3 days later.

### 2.4. Molecular Biology and Generation of Adenoviruses

pGW1CMV-Perceval plasmid, a gift from Gary Yellen (Addgene, Watertown, MA USA, plasmid #49082), was cloned into pShuttleCMV by UCLA Vector Core as described by the Rutter’s group [13]. Adenoviral particles were produced by the AdEasy Adenoviral Vector System (Agilent, Santa Clara, CA, USA. #240009) by transfecting HEK293T cells. The medium was changed 24 h after transfection. Adenoviruses were harvested at 48 h and 72 h after transfection. Viral supernatants were filtered with 0.45-μm filters and stored at –80 °C.

### 2.5. FLIM Imaging

To monitor the NAD(P)H metabolism change, FLIM imaging was conducted using a Leica SP8 DIVE FALCON laser scanning microscope equipped with a 63x/1.2NA water immersion objective for acquiring images of dispersed islet cells. NAD(P)H was detected at a 2-photon excitation wavelength of 740 nm and an emission wavelength of 440 to 500 nm. Live cell FLIM imaging was performed right before and 5 min after glucose stimulation in 256 × 256-pixel format at 0.4 frames per sec, with 6 to 8 frame repeats to reach 100 photons/pixel in the cytoplasm. Cells were fixed by 4% PFA right after imaging for further identification of cell type by immunostaining, performed as described previously by Wang et al. [5].

### 2.6. Organelle Live Staining

The microtubule and actin were stained by live cell dye SiR-tubulin (Cytoskeleton, Inc. Denver, CO, USA. CY-SC002) and SiR-actin (Cytoskeleton, Inc. Denver, CO, USA. CY-SC001), respectively, at 1 μM in TCM, with addition of 10 μM verapamil, at 37 °C, 5% CO_2_. INS832/13 cells and primary beta cells were stained for 30 min and 4 h, respectively, before imaging or stimulation. For co-staining with cytoskeleton, mitochondria were labeled by abberior LIVE ORANGE (abberior GmbH, Göttingen, Germany. LVORANGE) for 30 min in INS832/13 cells and 1 h in primary beta cells at 37 °C, 5% CO_2_.

### 2.7. Glucose Stimulation

Dispersed islet cells were starved in 4 mM glucose KRBH buffer (pH 7.4, 111 mM NaCl, 25 mM NaHCO_3_, 4.8 mM KCl, 1.2 mM KH_2_PO_4_, 1.2 mM MgSO_4_, 10 mM HEPES, 2.3 mM CaCl_2_ and 0.1% bovine serum albumin (BSA)) for 30 min in the microscope incubator and then treated with 16 mM glucose KRBH buffer for 30 min.

### 2.8. ATP and Calcium Live Imaging

To visualize ATP and calcium levels within cells, ATP was labeled using the Perceval HR fluorescent sensor, and calcium was labeled using Fura Red dye (Invitrogen™, Thermo Fisher Scientific Inc, Irwindale, CA, USA. Cat#F3020). Imaging was performed on a Confocal SP8-STED or Zeiss 880 microscope equipped with a 63x/1.20 water immersion objective lens. Perceval HR was excited at 488 nm with emission collected between 495 and 535 nm, while Fura Red was excited at 488 nm with emission collected between 620 and 660 nm. The pinhole was set to 2 Airy Units (AU), and images were acquired at 5 s intervals.

Fluorescence intensity data were background-thresholded, analyzed, and presented as F/F0 for Perceval and F0nuclear/F for Fura Red. Two circular regions of interest (ROIs) with a 2 μm diameter were selected at the submembrane or perinuclear sides along the longest polar axis of the cell cytoplasm for subcellular analysis. In the ATP signal analysis of Perceval, F0 represents the average fluorescence intensity of 10 time points recorded right before 16 mM glucose stimulation at submembrane ROI or perinuclear ROI. ATP fold change was shown as normalized fluorescence intensity ratio F/F0, where F was the fluorescence intensity of the same ROI at different time points. In the calcium signal analysis of Fura-Red, F0nuclear represents the average fluorescence intensity of 10 time points before glucose stimulation at the nuclear region. Cytoplasmic [Ca^2+^] fold change was shown as normalized fluorescence intensity ratio F0nuclear/F, where F was the fluorescence intensity of the submembrane ROI or perinuclear ROI at different time points. Cytoplasmic [Ca^2+^] recording was verified in INS832/13 cells exposed to solutions containing 10 μM ionomycin under conditions of “Ca^2+^-free” (0.5 mM EGTA) and “Ca^2+^-max” (5 mM Ca^2+)^ (Biotium, Fremont, CA, USA, Cat#59100). Cytoplasmic [Ca^2+^] changes were observed within the linear range, while nuclear [Ca^2+^] approached an intensity plateau.

### 2.9. Quantification and Statistical Analysis

The results are presented as mean ± SD. Wilcoxon signed-rank test was used in FLIM analysis. Significance was assumed at *p* < 0.05. Calcium and ATP traces were smoothed using a 3-point moving average.

## 3. Results

### 3.1. Mitochondria Are Present in the Submembrane Zone of Beta Cells

For the proposed hypothesis to hold, there must be mitochondria present in the submembrane zone of beta cells comparable, for example, to those described in neuronal synapses [14,15].

We first examined the distribution of mitochondria in INS832/13 cells, as their flattened shape permits high-resolution microscopy of the submembrane zone. We highlighted cell boundaries using a live-cell dye specific for tubulin or actin and used fluorescent stains to preferentially label the mitochondria (Figure 1A–D). Confocal microscopy revealed that mitochondria (pseudo-colored green) were invariably present in the submembrane region. In some optical sections, they were seen closely aligned with invaginations of the submembrane cytoskeleton, reminiscent of the tethers described previously in neurons (Figure 1B,D arrowheads). Similar imaging of primary mouse beta cells (Figure 1E,G) confirmed the presence of mitochondria in the submembrane zone; again, some optical sections revealed close associations with the submembrane cytoskeleton (Figure 1F,H arrowheads).

### 3.2. Preferential Early Activation of Mitochondrial OxPhos in the Submembrane Zone of Beta Cells in Response to a Glucose Stimulus

We extended the above colocalization studies by asking if glucose stimulation preferentially activated OxPhos in mitochondria in the submembrane zone, as predicted by our hypothesis. FLIM offers a direct means to assay the metabolism in different regions of the cell, making it straightforward to compare the regional increases in OxPhos in primary living beta cells upon exposure to a glucose stimulus.

While the basal metabolic status varied between isolated beta cells, 5 min after a glucose stimulus, OxPhos was increased to a greater extent in the submembrane zone than in the perinuclear zone (Figure 2, Appendix A). In beta cells with the most prominent response to glucose stimulation, there was a clear early response of the mitochondria that were localized to the submembrane zone (Figure 2A). This finding was further validated in studies that combined monitoring the beta cell response to glucose stimulation by FLIM combined with the visualization of the cellular cytoskeleton by SIR-tubulin dye (Figure 3). Three-dimensional reconstruction of these cell images allowed the detailed visualization of the close association between submembrane regions of enhanced OxPhos and the cellular submembrane cytoskeleton, and in some cases, adjacent to invaginations of the cytoskeleton (Figure 3, arrowhead).

### 3.3. Subcellular Ca^2+^ and ATP Responses to a Glucose Stimulus

To further explore our hypothesis, we simultaneously imaged the submembrane and perinuclear ATP and Ca^2+^ levels in beta cells following an increase in glucose as an extension to the above FLIM study that showed regional activation of mitochondria. ATP and Ca^2+^ were visualized by adenovirus-transfected Perceval ATP reporter and Fura Red dye, respectively, in live beta cells. There was considerable cell-to-cell variability observed for dispersed beta cells, as noted earlier, that may reflect beta cell functional heterogeneity or different degrees of injury and/or cell loss of function between cells following islet isolation and dispersion to single cells (Figure 4 and Appendix A).

Nonetheless, in some beta cells, glucose stimulation resulted in a prompt increment in the submembrane ATP concentration (as measured using Perceval [13]) and amplification of ~4 min period oscillations (Figure 4). In the most glucose-responsive beta cells, the prompt increase in submembrane ATP was mirrored by locally high Ca^2+^ levels, and these also had ~4 min oscillations.

Having established that a glucose challenge initially activates submembrane mitochondria to OxPhos and that there is a locally increased ATP produced at this site, we turned to electron microscopy of the mouse pancreas to determine whether submembrane mitochondria are present in relation to capillary delivery of glucose.

### 3.4. Beta Cell Mitochondria Are Proximate to the Islet Capillary Interface

We examined islet sections from mice using transmission electron microscopy, focusing on areas where the interface between beta cells and adjacent islet capillaries was in the plane of the section (Figure 5).

Consistent with prior studies carried out in neurons, we identified mitochondria near the cell boundary and identified mitochondria with densities that appear to tether them to the cell membrane opposite the beta cell capillary interface (Figure 5 arrowheads).

Some reports have emphasized the mobility of mitochondria, usually in transformed cell lines rather than primary beta cells, raising the question of whether the selective activation of submembrane mitochondria represents the activation of local sensory specialist mitochondria or perhaps a migration of mitochondria to the membrane in response to stimulatory glucose [16]. To address this, we imaged mitochondria in living INS832/13 cells following glucose stimulation (Appendix A). The mitochondrial area in the submembrane and perinuclear regions was quantified before and after glucose stimulation in 11 live cells at 22 locations using the non-potential-dependent dye, MitoID-Red. No significant changes were observed in the mitochondrial area relative to the total cytoplasmic area. We observed no net migration of mitochondria to the submembrane region upon glucose exposure.

We conclude that there are submembrane mitochondria present at the interface between beta cells and islet capillary endothelium in beta cells that potentially serve as early glucose sensors to integrate the detection of an increment in blood glucose with prompt insulin secretion.

## 4. Discussion

The present studies are consistent with the hypothesis that there is compartmentalization of the coupling of glucose sensing with regulated insulin secretion by pancreatic beta cells. We identified several key functional components of the glucose-induced insulin secretion system present and preferentially responsive to an increment in glucose in the submembrane region of beta cells.

Mitochondria are present in the beta cell submembrane region, and in some sections, they are apparently tethered to the submembrane cytoskeleton, as previously reported for neuronal synapses [14]. Synapses and beta cells share tightly regulated rapid discharge of secretory granules in response to an electrical stimulus. For beta cells, this electrical stimulus is initiated by membrane depolarization, which is accomplished by the inhibition of K_ATP_ channels by ATP. We observed an early glucose-induced activation of submembrane mitochondria in OxPhos and increased submembrane ATP/ADP concentrations. Notably, the submembrane ATP response was in the form of amplified ~4 min oscillations that were mirrored by the amplification of submembrane Ca^2+^ oscillations, as reported previously by Li et al. in mouse and human beta cells in elegant studies applying Fura–Red dye combined with TIRF microscopy [17]. Though not directly measured in the present study, insulin is secreted in ~4 min pulses that are amplified in response to hyperglycemia [18,19], consistent with these findings and the proposed model (Figure 5D).

Models have been advanced for the origin of the pulsatile secretion of insulin [20,21]. In general, these models depend on cyclical feedback on the proximal steps of the signal system. The submembrane localization of the signal response system adds credibility to those models. For example, while ATP triggers membrane depolarization and Ca^2+^ influx, the repolarization of the cell membrane consumes ATP. High submembrane Ca^2+^ following membrane depolarization would be taken up by submembrane mitochondria and drive another rise in ATP production as the key enzymes of OxPhos are Ca^2+^ dependent. Griesche et al. reported that submembrane mitochondria in the MIN6 beta cell line take up Ca^2+^ following glucose stimulation, consistent with the proposed model [22].

There are several important gaps in knowledge that have yet to be filled to establish the proposed model. The composition and nature of the proposed submembrane mitochondrial tethers are yet to be characterized for either neuronal synapses or beta cells. Also, the observed heterogeneity between isolated beta cells in these studies might reflect functional heterogeneity between beta cells in vivo or different degrees of damage resulting from the isolation of islets and dispersal to single cells. Functional and genetic heterogeneity has previously been reported in isolated beta cells, but these studies are also vulnerable to the same experimental limitation [23]. It is plausible that there are specialist beta cells engaged in coupling glucose concentrations with insulin secretion since, in pancreatic islets, beta cells are electrically coupled, and so only a subset of beta cells would need to subserve this role. It would require in vivo studies of vascularized islets with single-cell resolution to resolve the question, is there a subset of beta cells that serve to couple glucose sensing with insulin secretion? The potential role of PEP and other metabolic signals, which has been a recent debate [24], cannot be addressed using FLIM and requires further investigation employing complementary approaches. Another limitation of the current study is that it does not address whether the proposed subcellular specialist zones coupling glucose sensing with insulin secretion are disrupted in diabetes. While we are confident in the validity and conservative interpretation of our data—as we used complementary approaches to measure different types of cellular signals—future work will incorporate mitochondrial calcium sensors, as recently described by Jeyarajan S. et al. [25], to further validate these findings. Future investigations will be necessary to determine if this focal coupling is impaired in various forms of diabetes, which could provide critical insights into disease development.

## 5. Conclusions

In both type 1 and 2 diabetes, there is a progressive decline in glucose-mediated insulin secretion characterized by the attenuation of the magnitude of insulin pulses [26,27]. A fuller understanding of the subcellular organization of the coupling of glucose sensing with insulin secretion might be instructive for the underlying basis of impaired glucose-mediated insulin secretion in type 1 and 2 diabetes. Also, there is a delay in the glucose responsiveness of stem cell-derived human islets following engraftment [28]**.** It would be interesting to establish the time course of development of the stem cell-derived beta cell submembrane regional functional architecture and its relation to the recipient vascular endothelium.

## Figures and Tables

**Figure 1 cells-14-00198-f001:**
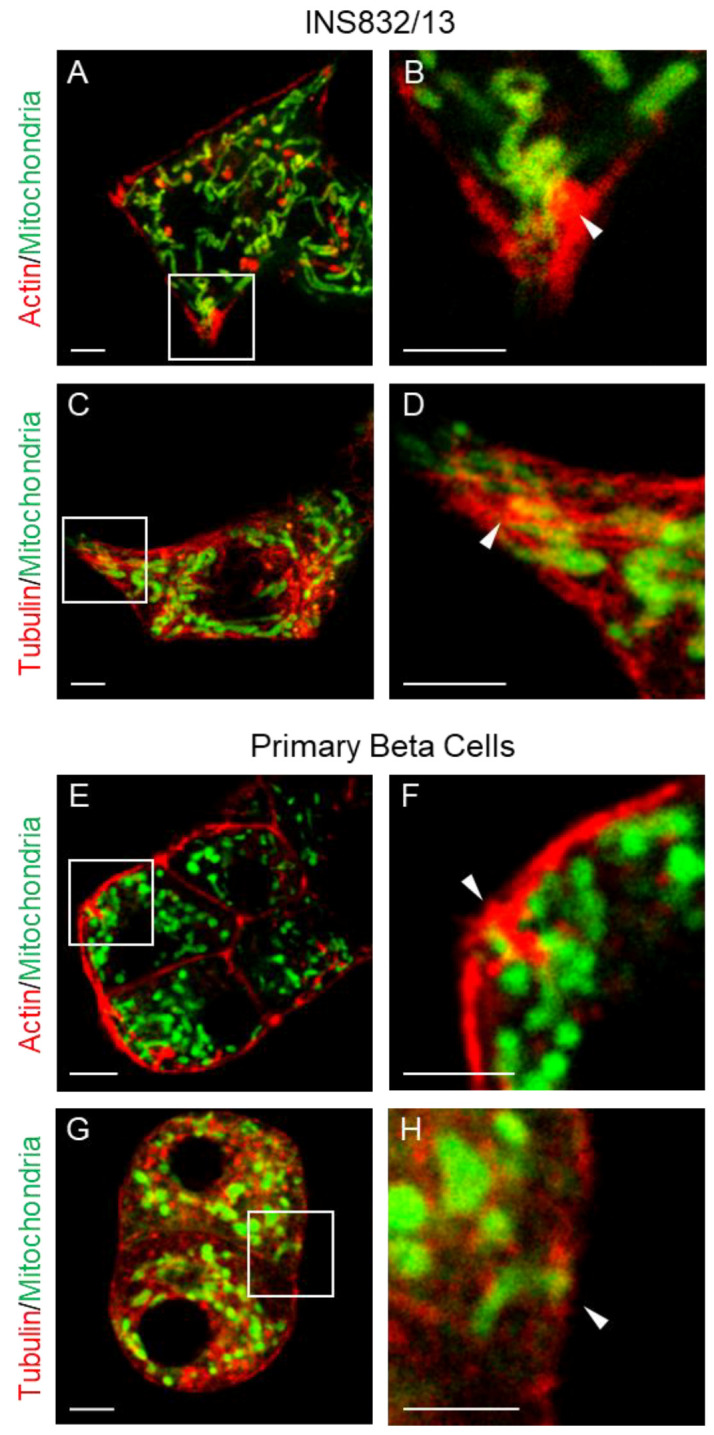
Mitochondria in the submembrane zone of beta cells. We outlined the cytoskeleton in live INS823/13 beta cells (**A**–**D**) and primary mouse beta cells (**E**–**H**) with live dye for either actin or tubulin followed by abberior LIVE ORANGE mito staining to visualize the mitochondria. Using Leica SP8 confocal microscopy with an 86x/1.20 W objective, mitochondria were invariably present in the submembrane region. In some sections, there was a close interaction between mitochondria and protrusions of the submembrane cytoskeleton (white arrow in inset (**B**,**D**,**F**,**H**), corresponding to the white box in panel (**A**,**C**,**E**,**G**)), which was reminiscent of the mitochondrial tethering previously reported in neurons. Scale bar = 5 μm.

**Figure 2 cells-14-00198-f002:**
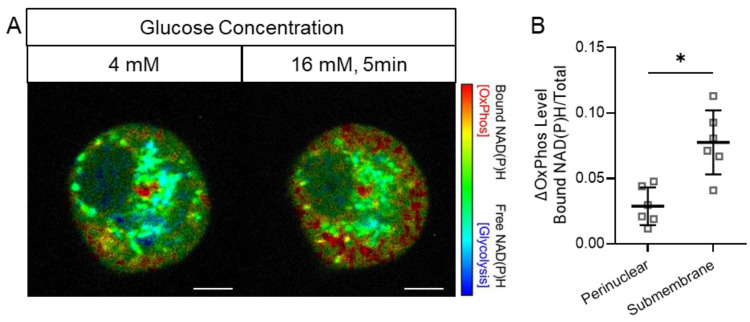
Application of FLIM in beta cells following a glucose challenge reveals an early preferential metabolic trajectory to OxPhos in submembrane mitochondria. FLIM imaging of a representative primary mouse beta cell (**A**) at basal glucose concentration (4 mM) and 5 min after glucose stimulation (16 mM). Following glucose stimulation, there is a predominant transition to oxidative phosphorylation in the submembrane zone, color-coded as red, depicting increased bound/total NAD(P)H). The quantification of the change in the ratio of bound/total NAD(P)H by FLIM in the perinuclear versus the submembrane zone (**B**) is provided in 6 primary mouse beta cells 5 min after glucose stimulation. NAD(P)H FLIM signal was captured by 2-photon excitation at 740 nm and emission between 440 and 500 nm. Data are presented as mean ± SD; * *p* < 0.05, Wilcoxon signed-rank test. Scale bar, 5 μm. Beta cells (n = 6) were from dispersed mouse islets obtained from three independent isolations.

**Figure 3 cells-14-00198-f003:**
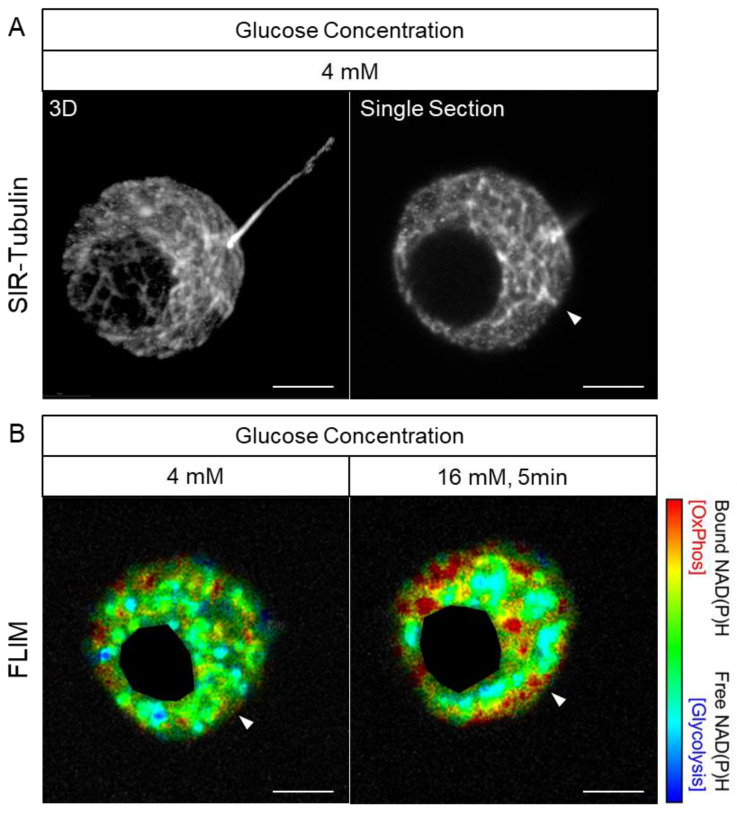
Early submembrane activation of oxidative phosphorylation is associated with apparent tethered mitochondria to the tubulin network. Microtubule network and a cilium are visualized in stacked Z plane images (**A**) of SIR-tubulin dye decorated representative primary mouse beta cell. In a single plane of section (**B**) of the same cell, FLIM imaging at basal glucose (4 mM) and 5 min after exposure to glucose stimulation (16 mM) reveals an early transition to OxPhos in a presumed mitochondrion closely associated with a tubulin enriched region (white arrowhead) consistent with early activation of submembrane tethered mitochondria. A cilium is identified (A) in the stacked images, and in this cell, there is no suggestion of early activation of OxPhos at the base of the cilium in response to glucose stimulation. Scale bar, 5 μm.

**Figure 4 cells-14-00198-f004:**
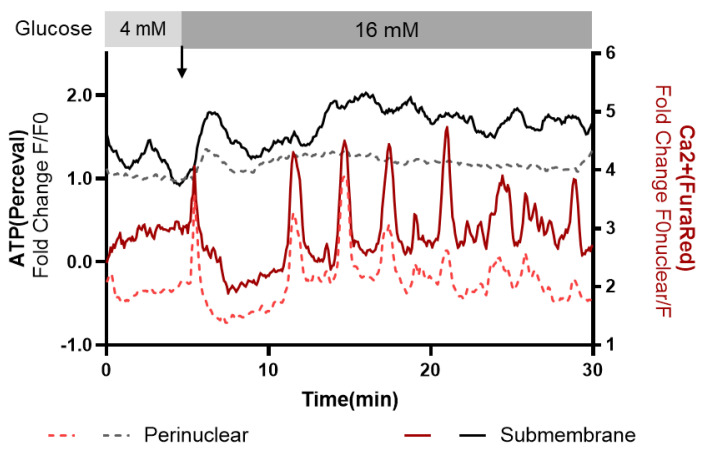
Submembrane Ca^2+^ and ATP responses to glucose stimulation in isolated beta cells. We monitored the relative submembrane and perinuclear levels of Ca^2+^ and ATP in isolated primary beta cells at basal glucose (4 mM) and following glucose stimulation (16 mM). There was a wide variance between individual beta cells. In the more responsive cells, exemplified in Figure 4, there were clear ~4 min oscillations of both Ca^2+^ and ATP that were promptly amplified in response to glucose stimulation predominantly in the submembrane zone. The arrow indicates the addition of glucose. Changes in ATP levels were measured using a Perceval-HR sensor (black trace), and changes in Ca^2+^ level were measured using Fura Red (red trace).

**Figure 5 cells-14-00198-f005:**
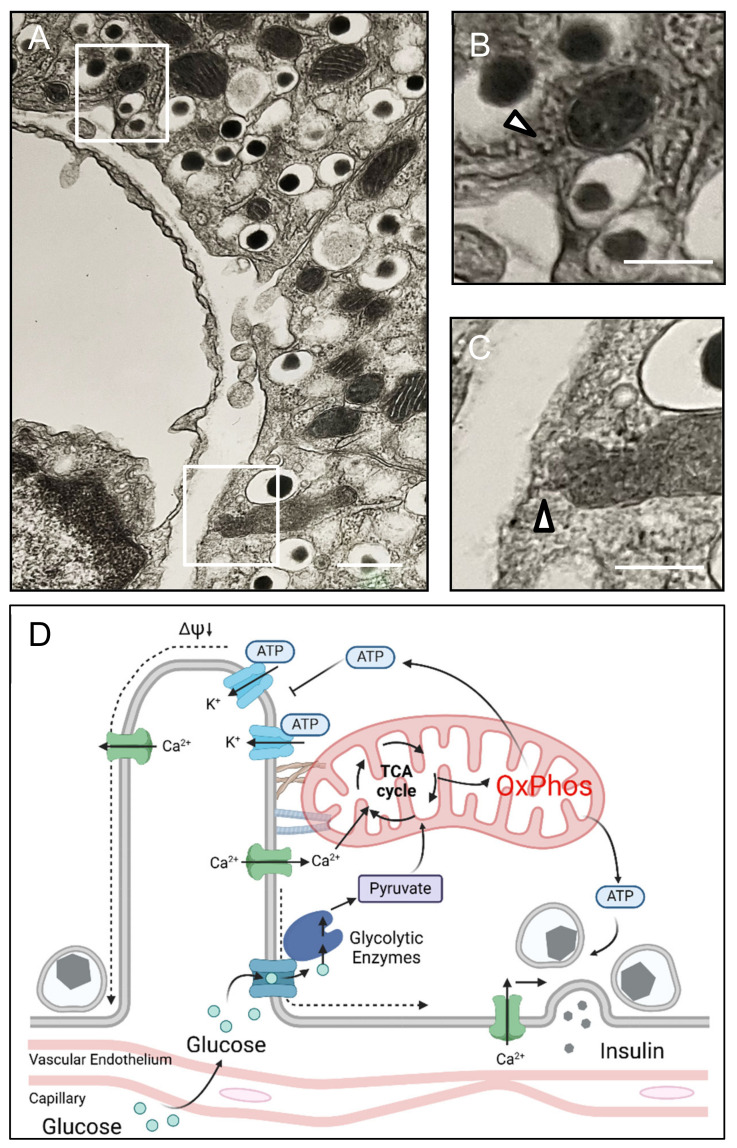
Beta cell capillary relationship and proposed revised glucose sensing insulin secretion model. To evaluate the putative submembrane mitochondria and vascular endothelium of adjacent capillaries in beta cells, we deployed transmission electron micrography of fixed mouse whole pancreas and identified sections that included sections of capillaries ((**A**), scale bar 500 nm; white box indicate the region displayed in (**B**,**C**)). In some sections of beta cells facing capillaries, submembrane mitochondria were closely affiliated with apparent membrane-associated tethers (**B**,**C**; scale bar 250 nm) consistent with prior studies of neuronal synapses. Based on the studies presented here, as well as studies published elsewhere, we propose a revision of the canonical model of beta cell glucose sensing coupled to insulin secretion (**D**). We propose that, at least in some beta cells, there is a microdomain at the capillary endothelium interface where glucose delivered by the adjacent capillary is metabolized to pyruvate in a concentration-dependent manner. Pyruvate is then accessed by submembrane tethered mitochondria to produce ATP that is delivered directly to local K_ATP_ channels in the cell membrane. This prompts depolarization of the cell membrane and an influx of Ca^2+^ to the submembrane zone, prompting exocytosis of primed docked insulin secretory vesicles. The locally high submembrane Ca^2+^ is then taken up by the membrane-adjacent mitochondria, driving the Ca^2+^-dependent OxPhos enzymes to generate the next pulse of ATP. The latter is facilitated by the availability of ADP as ATP is consumed by repolarizing the cell membrane and priming and docking the next repertoire of insulin secretory vesicles to the cell membrane.

## Data Availability

The data supporting the findings of this study are available from the corresponding author upon reasonable request.

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
