# Peer review of "Subcellular Compartmentalization of Glucose Mediated Insulin Secretion"

_cells, 2025, doi:10.3390/cells14030198_

Round 1
Reviewer 1 Report
Comments and Suggestions for Authors
cells-3380020
Subcellular Compartmentalization of Glucose Mediated Insulin Secretion
The authors of the paper presented the cellular response of the pancreatic beta cells after glucose stimulation, showing the first action in beta cells related to glucose sensing, mitochondria localization, activation, and further signal transduction. Most of the results are based on microscopic analysis.
This is a good paper. Results are presented in a good and clear way.
The goals and hypothesis of the study are clearly presented in the last part of the introduction. The introduction section contains sufficient information to understand the topic of the study.
There are a few comments mostly related to the description of the methods:
In the material and method section, some protocols (eg. line 93, but not only) are not described but references are given. In my opinion, it would be good to briefly describe the protocol anyway.
ATP and calcium calculation is not well described and in my opinion, might be improved for better understanding by the readers.
The results presented in the paper correspond to the general knowledge and give further insight into detailed information related to beta cell action and insulin regulation of insulin secretion.
Author Response
We appreciate the constructive suggestions of the reviewers and provide our responses below the comments and suggestions for each reviewer.
Reviewer 1:
Comment 1:
The authors of the paper presented the cellular response of the pancreatic beta cells after glucose stimulation, showing the first action in beta cells related to glucose sensing, mitochondria localization, activation, and further signal transduction. Most of the results are based on microscopic analysis.
This is a good paper. Results are presented in a good and clear way.
The goals and hypothesis of the study are clearly presented in the last part of the introduction. The introduction section contains sufficient information to understand the topic of the study.
Response 1:
Thank you for carefully reading the paper and your supportive comments,
Comment 2:
There are a few comments mostly related to the description of the methods:
In the material and method section, some protocols (eg. line 93, but not only) are not described but references are given. In my opinion, it would be good to briefly describe the protocol anyway.
Response 2:
Thank you for this suggestion. As requested we have added more details on islet isolation: now line 92, “…with collagenase for 17 min in 37℃ water bath and further handpicked as…”; slide coating: now line 100, “…for 24 h at 4℃ overnight…”; immunostaining: now line 117, “…Cells were fixed by 4% PFA right after imaging…”
Comment 3:
ATP and calcium calculation is not well described and in my opinion, might be improved for better understanding by the readers.
Response 3:
Thank you for the suggestion. The studies undertaken to validate the ATP and Ca data presented in the manuscript are now more fully included in the methods section. Line 142-152, “Two circular regions of interest (ROIs) with a 2 μm diameter were selected at the sub-membrane or peri-nuclear sides along the longest polar axis of the cell cytoplasm for subcellular analysis. In ATP signal analysis of Perceval, F0 represents the average fluorescence intensity of 10 time points at 5-second intervals recorded right before 16 mM glucose stimulation at submembrane ROI or peri-nuclear ROI. ATP fold change was shown as normalized fluorescence intensity ratio F/F0, where F was the fluorescence intensity of the same ROI at different time points. In Calcium signal analysis of Fura-Red, F0 nuclear represents the average fluorescence intensity of 10 time points at 5-second intervals before glucose stimulation at the nuclear region. Cytoplasmic [Ca²⁺] fold change was shown as normalized fluorescence intensity ratio F0 nuclear/F, where F was the fluorescence intensity of the submembrane ROI or peri-nuclear ROI at different time points.”
Comment 4:
The results presented in the paper correspond to the general knowledge and give further insight into detailed information related to beta cell action and insulin regulation of insulin secretion.
Response 4:
Thank you.

Reviewer 2 Report
Comments and Suggestions for Authors
In this well written and thought-provoking study the Authors propose a new model for glucose sensing coupling to insulin secretion in pancreatic beta-cells. It would be interesting to investigate if the mechanism is impaired by diabetes/hyperglycemia; a proof-of-concept experiment with INS1 beta-cell line may be feasible and will strengthen the significance of the reported results. In addition, I would really like to see the present results confirmed in human islets, however I do understand that the effort required may be disproportionated to the present publication; nonetheless, I suggest that the Authors acknowledge this limit in the discussion section.
Author Response
We appreciate the constructive suggestions of the reviewers and provide our responses below the comments and suggestions for each reviewer.
Reviewer 2:
Comment 1:
In this well written and thought-provoking study the Authors propose a new model for glucose sensing coupling to insulin secretion in pancreatic beta-cells. It would be interesting to investigate if the mechanism is impaired by diabetes/hyperglycemia; a proof-of-concept experiment with INS1 beta-cell line may be feasible and will strengthen the significance of the reported results. In addition, I would really like to see the present results confirmed in human islets, however I do understand that the effort required may be disproportionated to the present publication; nonetheless, I suggest that the Authors acknowledge this limit in the discussion section.
Response 1:
Thank you for these kind comments, we greatly appreciate your careful reading of the manuscript and for noting the escalation from cell lines to primary beta cells. We agree that the next step is to establish whether the proposed subcellular specialist zone coupling glucose sensing with insulin secretion is disrupted in diabetes. As suggested, we not only note that it will be of interest to establish if this focal coupling of glucose sensing with insulin secretion is disrupted in various forms of diabetes, and that a limitation of the present study is that we have not yet addressed this.
Now line 339-343, “A limitation of the present study is that it does not address whether the proposed subcellular specialist zones coupling glucose sensing with insulin secretion are disrupted in diabetes. Future investigations will be necessary to determine if this focal coupling is impaired in various forms of diabetes, which could provide critical insights into disease development”

Reviewer 3 Report
Comments and Suggestions for Authors
The work from Wang et al, titled “Subcellular compartmentalization of glucose-mediated insulin secretion”. Aim to study local mitochondrial metabolic responses upon glucose stimulation in pancreatic cells. To study the subcellular responses, the authors used quantitative measurements of NADPH using FLIM combined with epifluorescence imaging of ATP levels and Ca2+. The authors described that upon glucose stimulation, pancreatic cells show a differential response by activating the OXPHOS in mitochondria close to the subplasmalemmal region. The authors show that the glucose-responsive cells display local ATP increases synchronized with Ca2+ oscillations. The authors attribute these changes in the ATP to the localization of subplasmalemmal mitochondria. Overall, the paper is well written and pursues to study a topic in the field which is technically very challenging as it is the local differential responses of mitochondria. However, this reviewer considers that the statements do not support the conclusions, and more controls are required to have a clear picture of the subplasmalemmal responses. Here I summarize the major comments
1. The main concern from the FLIM experiments is that is not clear that there is a local subcellular response to the increase in the glucose, actually it looks like mitochondria respond in a heterogenous manner but it doesn’t show a clear relationship with the positioning of the mitochondria. One way to address this could be to measure simultaneously
2. Please confirm with a mitochondrial calcium sensor that the local cytosolic Calcium oscillations are translated in mitochondrial Ca2+ responses in the subplasmalemmal responses.
3. Also, the ATP changes measured by Perceval can also suggest that the ATP generated is from local glycolysis and not from mitochondria. Pleased add proper controls such as mitochondrial poisons and direct mitochondria measurements with mito-Perceval or mt-Atea,.
4. One of my main concern are the measurements with Perceval, which is highly sensitive to pH changes, and local membrane depolarization can cause local pH changes due to ion exchange. I think that it would be beneficial to control the pH changes with a sensor.
5. To date there is no PM-mitochondria molecular tether identified in mammalian cells, as it has been identified for yeast, and most of the mitochondria found in close apposition to the plasma membrane are usually associated with an ER cisternae. Also, the authors should quantify the number of mitochondria in the subplasmalemmal area that has this type of contact. Since the study lacks the resolution to address a physical tethering between the PM and mitochondria, I would downsize the observation in the conclusion and put more emphasis on how mitochondria can respond to local signals. If the authors would like to pursue the PM-mitochondria interaction, then measurements with a mitochondria-PM linker are required to confirm the findings
6. Finally, Insulin secretion is mentioned in the title but was not measured
Author Response
We appreciate the constructive suggestions of the reviewers and provide our responses below the comments and suggestions for each reviewer.
Reviewer 3:
Comment 1:
The work from Wang et al, titled “Subcellular compartmentalization of glucose-mediated insulin secretion”. Aim to study local mitochondrial metabolic responses upon glucose stimulation in pancreatic cells. To study the subcellular responses, the authors used quantitative measurements of NADPH using FLIM combined with epifluorescence imaging of ATP levels and Ca2+. The authors described that upon glucose stimulation, pancreatic cells show a differential response by activating the OXPHOS in mitochondria close to the subplasmalemmal region. The authors show that the glucose-responsive cells display local ATP increases synchronized with Ca2+ oscillations. The authors attribute these changes in the ATP to the localization of subplasmalemmal mitochondria. Overall, the paper is well written and pursues to study a topic in the field which is technically very challenging as it is the local differential responses of mitochondria. However, this reviewer considers that the statements do not support the conclusions, and more controls are required to have a clear picture of the subplasmalemmal responses. Here I summarize the major comments
Response 1:
Thank you for carefully reading the manuscript and your suggestions. We address each of them below.
Comment 2:
1). The main concern from the FLIM experiments is that is not clear that there is a local subcellular response to the increase in the glucose, actually it looks like mitochondria respond in a heterogenous manner but it doesn’t show a clear relationship with the positioning of the mitochondria. One way to address this could be to measure simultaneously
Response 2:
Thank you for this suggestion. We previously validated the application of FLIM under these conditions in live beta cells to correctly identify the role of mitochondria (Wang Z, et al., Communications biology, 2021). We specifically established there was overlap between mitochondrial position with oxidative phosphorylation (OXPHOS) activity under similar experimental conditions. That study demonstrated there was a consistent spatial correlation between mitochondria and functional OXPHOS activity. For clarity we have added this point to the methods section citing the relevant paper. Now line 117-119: We previously validated the correct alignment of the FLIM OxPhos signal with the cellular location of mitochondria in living beta cells under the conditions applied in the current study [5].
Comment 3:
- Please confirm with a mitochondrial calcium sensor that the local cytosolic Calcium oscillations are translated in mitochondrial Ca2+ responses in the subplasmalemmal responses.
Response 3:
We did not use a specific mitochondrial Ca2+ probe in these studies as this was not the key question addressed here. Our focus was instead on the submembrane Ca2+ response as described in the manuscript. We agree it would be of interest to pursue further studies of subcellular regional mitochondria, particularly in the context of type 2 diabetes where a key component of cellular dysfunction is aberrant Ca2+signaling, most likely from toxic oligomer induced ER Ca2+ leaks but as per reviewer 2 our focus here was exploring the concept of subcellular coupling of glucose sensing rather than on dysfunction in diabetes.
Comment 4:
- Also, the ATP changes measured by Perceval can also suggest that the ATP generated is from local glycolysis and not from mitochondria. Pleased add proper controls such as mitochondrial poisons and direct mitochondria measurements with mito-Perceval or mt-Atea,.
Response 4:
This question appears to arise from the controversial and widely contested hypothesis of glucose insulin secretion of Merrins and colleagues (see: Satin LS et al, Diabetes 2024:73(6):844-848). We respectfully suggest that the onus of responsibility to validate that hypothesis lies with Merrins and colleagues.
Comment 5:
- One of my main concern are the measurements with Perceval, which is highly sensitive to pH changes, and local membrane depolarization can cause local pH changes due to ion exchange. I think that it would be beneficial to control the pH changes with a sensor.
Response 5:
We thank you for this suggestion but are unaware of any data to suggest a local pH gradient exists in the submembrane region of the intracellular space as it is strongly buffered. Furthermore, as noted in the manuscript, the observation of submembrane locally high ATP is not new, as we cited (Tarasov AI, et al., PLoS One. 2012).
Comment 6:
- To date there is no PM-mitochondria molecular tether identified in mammalian cells, as it has been identified for yeast, and most of the mitochondria found in close apposition to the plasma membrane are usually associated with an ER cisternae. Also, the authors should quantify the number of mitochondria in the subplasmalemmal area that has this type of contact. Since the study lacks the resolution to address a physical tethering between the PM and mitochondria, I would downsize the observation in the conclusion and put more emphasis on how mitochondria can respond to local signals. If the authors would like to pursue the PM-mitochondria interaction, then measurements with a mitochondria-PM linker are required to confirm the findings
Response 6:
We agree that the tethering has been shown in yeast and in neuronal synapses, as we cited (Perkins GA, et al., J Neurosci, 2010), a special cell region that is analogous to the proposed specialist zone we reported in beta cells.
Comment 7:
- Finally, Insulin secretion is mentioned in the title but was not measured
Response 7:
It was not clear to us whether or not the reviewer was contesting that an increase in glucose concentration results in beta cell insulin secretion? It is true that in these single cell experiments we did not directly measure insulin secretion. To address the reviewer’s comment, we now acknowledge this point in the discussion of the revised paper.
Now line 313, “Though not directly measure in the present study,…”

Round 2
Reviewer 3 Report
Comments and Suggestions for Authors
The paper from Wang et al has been revised previously. The authors only partially addressed this. Please see below my comments
1. This reviewer is still concerned about the subcellular distribution of the FLIM measurements, since the representative images doesn’t match with the conclusions. The images show an heterogenous inter-mitochondrial response upon HG and not a subplasmalemmal response. This reviewer asked for more controls to confirm the observations, however, the authors mentioned that they already validated the technique citing a previous paper from the group (Wang Z, et al., Communications biology, 2021). After revising the previous paper from the authors, the authors validated the FLIM measurements, however, this where not done at the organellar level, the validation was done at the tissue level and whole cell, which not necessary match with the scope of this study, since now authors are proposing a mechanism at the subcellular level. Again, I maintain my concerns that proper controls are missing.
2. Again, the authors attribute the increase in the lifetime of the NAD(P)H to Ca2+ oscillation and activation of the mitochondrial metabolism (which is also shown in the last scheme), however, the only measurement was done using Perceval, which is known to be highly dependent on pH changes. This reviewer suggested to measure mitochondrial Ca2+ and mitochondrial ATP, which could validate that the observed Ca2+ oscillations are translated into mitochondrial Ca2+ uptake and a subsequent local activation of the mitochondrial metabolism, which could explain that the increased in the ATP levels are coming from mitochondria and not local glycolysis. Additionally, the authors should find either another way to measure ATP or to use a pH sensor to validate that there are no changes in the pH.
Author Response
Comment 1: This reviewer is still concerned about the subcellular distribution of the FLIM measurements, since the representative images doesn’t match with the conclusions. The images show an heterogenous inter-mitochondrial response upon HG and not a subplasmalemmal response. This reviewer asked for more controls to confirm the observations, however, the authors mentioned that they already validated the technique citing a previous paper from the group (Wang Z, et al., Communications biology, 2021). After revising the previous paper from the authors, the authors validated the FLIM measurements, however, this where not done at the organellar level, the validation was done at the tissue level and whole cell, which not necessary match with the scope of this study, since now authors are proposing a mechanism at the subcellular level. Again, I maintain my concerns that proper controls are missing.
Response 1:
Thank you. While we utilized PercevalHR/Fura data in our manuscript, our key conclusions were in fact derived from the use of FLIM (Fluorescence Lifetime Imaging Microscopy). FLIM provides complementary and robust insights without requiring the use of any labeled probes, significantly strengthening our study’s validity.
Co-localization analysis of mitochondrial signals that we measured before and after elevating glucose concentration was already been addressed in prior studies conducted by our lab (Wang Z, et al., Communications biology, 2021). The validation of subcellular level colocalization was presented in Supplementary Figure 1b of the cited paper. This forms the basis of the conclusions we presented in Figure 2B.
While our current study does not include mitochondrial free calcium measurements, we plan to integrate mitochondrial calcium sensors in our future work, as described by Jeyarajan et al. (Biosensors, 2023). We are optimistic that these additional efforts will expand upon the findings presented here and will provide additional mechanistic insights.
To address reviewer’s concerns, we propose including the following additional clarifications in the revised manuscript line 314-345:
"While we are confident in the validity and conservative interpretation of our data—as we used complementary approaches to measure different types of cellular signals—future work will incorporate mitochondrial calcium sensors, as recently described by Jeyarajan S. et al. (Biosensors, 2023), to further validate these findings."
Comment 2:
Again, the authors attribute the increase in the lifetime of the NAD(P)H to Ca2+ oscillation and activation of the mitochondrial metabolism (which is also shown in the last scheme), however, the only measurement was done using Perceval, which is known to be highly dependent on pH changes. This reviewer suggested to measure mitochondrial Ca2+ and mitochondrial ATP, which could validate that the observed Ca2+ oscillations are translated into mitochondrial Ca2+ uptake and a subsequent local activation of the mitochondrial metabolism, which could explain that the increased in the ATP levels are coming from mitochondria and not local glycolysis. Additionally, the authors should find either another way to measure ATP or to use a pH sensor to validate that there are no changes in the pH.
Response 2:
Thank you. We understand the reviewer’s concerns regarding potential pH effects on the Perceval signals we measured. However, Li and Tengholm (Diabetologia, 2013) previously explored the impact of pH on Perceval measurements in their study of submembrane ATP in beta cells and successfully demonstrated that glucose-induced changes in ATP as detected with Perceval were not influenced by pH variations. A direct quote from their paper was that “The oscillations in Perceval fluorescence did not reflect changes in pH. Accordingly, glucose had little effect on pH in BCECF-loaded islets, while alkalinization with NH4Cl caused a dramatic rise of the BCECF and Perceval signals”. We believe that this applies as well to the very similar measurements we carried out in our study and thus strongly supports the robustness of our results.
Raising the concentration of glucose has been shown to clearly increase bulk cytosolic pH. In their seminal study of the Perceval probe which they designed, Berg et al (Nature Methods, 2009), showed that the baseline Perceval fluorescence was reduced at more acidic pH. However, despite the sensitivity of Perceval baseline fluorescence to pH, the magnitude of the changes that occurred in response to acute glucose challenges were unchanged in magnitude (see Fig 5b in Berg et al, 2009).